# A metabolic switch controls intestinal differentiation downstream of *Adenomatous* polyposis coli (APC)

Imelda T Sandoval[1], Richard Glenn C Delacruz[1], Braden N Miller[1], Shauna Hill[2,3], Kristofor A Olson[4], Ana E Gabriel[1], Kevin Boyd[1], Christeena Satterfield[1], Holly Van Remmen[2], Jared Rutter[4], David A Jones[1]*

[1]Functional and Chemical Genomics, Oklahoma Medical Research Foundation, Oklahoma City, United States; [2]Aging and Metabolism Research Program, Oklahoma Medical Research Foundation, Oklahoma City, United States; [3]Department of Cellular and Structural Biology, University of Texas Health Science Center at San Antonio, San Antonio, United States; [4]Department of Biochemistry, University of Utah School of Medicine, Salt Lake City, United States

**Abstract** Elucidating signaling pathways that regulate cellular metabolism is essential for a better understanding of normal development and tumorigenesis. Recent studies have shown that *mitochondrial pyruvate carrier 1 (MPC1)*, a crucial player in pyruvate metabolism, is downregulated in colon adenocarcinomas. Utilizing zebrafish to examine the genetic relationship between *MPC1* and *Adenomatous polyposis coli (APC)*, a key tumor suppressor in colorectal cancer, we found that *apc* controls the levels of *mpc1* and that knock down of *mpc1* recapitulates phenotypes of impaired *apc* function including failed intestinal differentiation. Exogenous human *MPC1 RNA* rescued failed intestinal differentiation in zebrafish models of *apc* deficiency. Our data demonstrate a novel role for *apc* in pyruvate metabolism and that pyruvate metabolism dictates intestinal cell fate and differentiation decisions downstream of *apc*.

*For correspondence: david-jones@omrf.org

**Competing interests:** The authors declare that no competing interests exist.

## Introduction

Mutations in the adenomatous polyposis coli (*APC*) gene are responsible for Familial Adenomatous Polyposis (FAP), a genetic predisposition to colorectal cancer, and are also found in the majority of sporadic colonic tumors (*Fearnhead et al., 2001*). Critical roles for APC in colon carcinogenesis are attributed to its ability to negatively regulate the proliferative consequences of Wnt signaling through degradation of $\beta$-catenin, and maintain normal intestinal differentiation by controlling the biosynthesis of retinoic acid (RA) (*Jette et al., 2004*; *Nadauld et al., 2006a*, *2004*, *2005*; *Rai et al., 2010*; *Schneikert et al., 2007*; *Shelton et al., 2006*). Although tremendous progress has been made in understanding the role of APC, its full battery of functions continue to expand.

Altered energy metabolism is an emerging hallmark in cancer (*Hanahan and Weinberg, 2011*). The observation that cancer cells produce energy to support cell growth and proliferation differently than normal cells is known as the Warburg effect, and refers to neoplastic cells favoring aerobic glycolysis, even in the presence of ample oxygen (*Vander Heiden et al., 2009*; *Warburg, 1956*). One of the major molecular mechanisms contributing to Warburg effect is mitochondrial dysfunction through impaired pyruvate metabolism (*Diaz-Ruiz et al., 2011*).

Pyruvate lies at the junction of glycolysis and the tricarboxylic acid (TCA) cycle. Contingent on the metabolic needs of the cell, pyruvate can be transported into the mitochondria, and through the action mainly of pyruvate dehydrogenase (PDH), it can be used to drive ATP production and

**eLife digest** Colon cancer remains an important problem in healthcare. Cancer researchers are looking for new ways to detect the disease earlier and treat it more effectively. This is challenging because many of the genetic and molecular causes of colon cancer are still poorly understood. Mutations in the gene that encodes a protein called APC are one of the major causes of the disease. The APC protein normally keeps cells from growing and dividing too fast or in an uncontrolled way and is hence referred to as a tumor suppressor. For example, APC induces stem cells in the intestine to develop into specialized cells that keep the gut working normally. Mutations in tumor suppressor genes are common in many cancers.

Other research has shown that cancer cells must reprogram their own metabolism – in other words, all the chemical processes that keep the cell alive – to meet the demands of proliferating rapidly. In particular, recent studies reveal that colon cancer cells produce less of a protein called mpc1, which is involved in metabolism. These discoveries raised the following questions: does APC have an additional role in maintaining normal metabolism in cells by controlling how much mpc1 is produced? Do mutations in the gene for APC lead to colon cancer because they alter the cell's metabolism?

Sandoval et al. have now discovered a connection between APC and changes in cancer cells that help them to adapt to a new metabolic program. Experiments with zebrafish – a model animal that is now commonly used in the field of cancer biology – showed that APC acts via mpc1 to regulate how the cell uses energy. This regulation goes awry in colon cells that have abnormal APC activity; however, restoring the cell's metabolism back to normal was enough to induce cells in the intestine to develop properly.

Together, these findings suggest that restoring the normal balance of energy production in colon cancer cells may be an effective way to make the cells behave normally. This hypothesis remains to be tested and, if confirmed, further studies will be needed to determine whether it will lead to new treatments for colon cancer in humans.

generate building blocks for macromolecule biosynthesis through oxidative phosphorylation. Alternatively, pyruvate can be converted to lactate via lactate dehydrogenase (LDH) and exported out of the cell. Aberrations in genes involved in pyruvate metabolism and transport have been reported in human diseases, particularly in cancer (*Gray et al., 2014*). For example, *monocarboxylate transporter 4 (MCT4)* and *LDHA* are overexpressed in cancer (*Kim et al., 2013*; *Rong et al., 2013*). An isoform of *pyruvate kinase 2 (PKM2)* is preferentially expressed in numerous cancer types including pancreatic, colon and lung, and has been shown to promote aerobic glycolysis in HeLa cells by functioning as a transcriptional coactivator for HIF-1 (*Cerwenka et al., 1999*; *Christofk et al., 2008*; *Luo and Semenza, 2011*; *Schneider et al., 2002*; *Yeh et al., 2008*). Restoration of the pyruvate dehydrogenase complex activity through inhibition of pyruvate dehydrogenase kinase 1 (PDK1) in head and neck squamous cell carcinoma cell lines led to reduced HIF-1a expression and tumor growth (*McFate et al., 2008*).

The recently identified mitochondrial pyruvate carrier subunit *MPC1* is part of the MPC complex that is responsible for the uptake of pyruvate into the inner mitochondrial matrix (*Bricker et al., 2012*; *Herzig et al., 2012*). Recent work has revealed that *MPC1* is downregulated in various human cancers and that this correlates with poor survival (*Schell et al., 2014*). Consistent with a causative role in tumorigenesis, re-expression of MPC1 repressed the Warburg effect in colon cancer cell lines (*Schell et al., 2014*). It is not clear how MPC1 is regulated or how its activities relate to the known genetic events that contribute to colon cancer development. Given the potential role for MPC1 in colorectal cancer and the importance of APC mutation, we investigated the mechanistic relationship between the mutational status of *apc* and *mpc1*. Herein, we report that *apc* regulates pyruvate metabolism by controlling the levels of *mpc1* via RA. Further, *mpc1* is required and sufficient for initiating normal intestinal differentiation downstream of *apc*. Our findings strongly suggest that changes in metabolic profile can drive cell fate and differentiation decisions.

## Results

### *mpc1* and *mpc2* are downregulated in *apc*-deficient zebrafish

To investigate the relationship between *apc* and *mpc*, we utilized the *apc^mcr^* zebrafish, which is homozygous for a truncating mutation in the Mutation Cluster Region (MCR) of *apc* and similar to what is found in human colon tumors (*Hurlstone et al., 2003*; *Miyoshi et al., 1992a*, *1992b*). In parallel, we also knocked down the expression of *apc* in wild type (WT) embryos using antisense morpholino (*apc* mo) (*Figure 1—figure supplement 1*). Evaluating gene expression of *mpc1* and *mpc2* by qRT-PCR, we found that both genes were significantly downregulated in *apc^mcr^* and *apc* mo embryos compared to WT/het siblings and control mo, respectively (*Figure 1A,B*). This was confirmed by whole mount in situ hybridization for *mpc1* and *mpc2* (*Figure 1C,D*). Additional in situ analyses for *mpc1* and *mpc2* in WT embryos revealed staining in the head, eyes, vasculature and

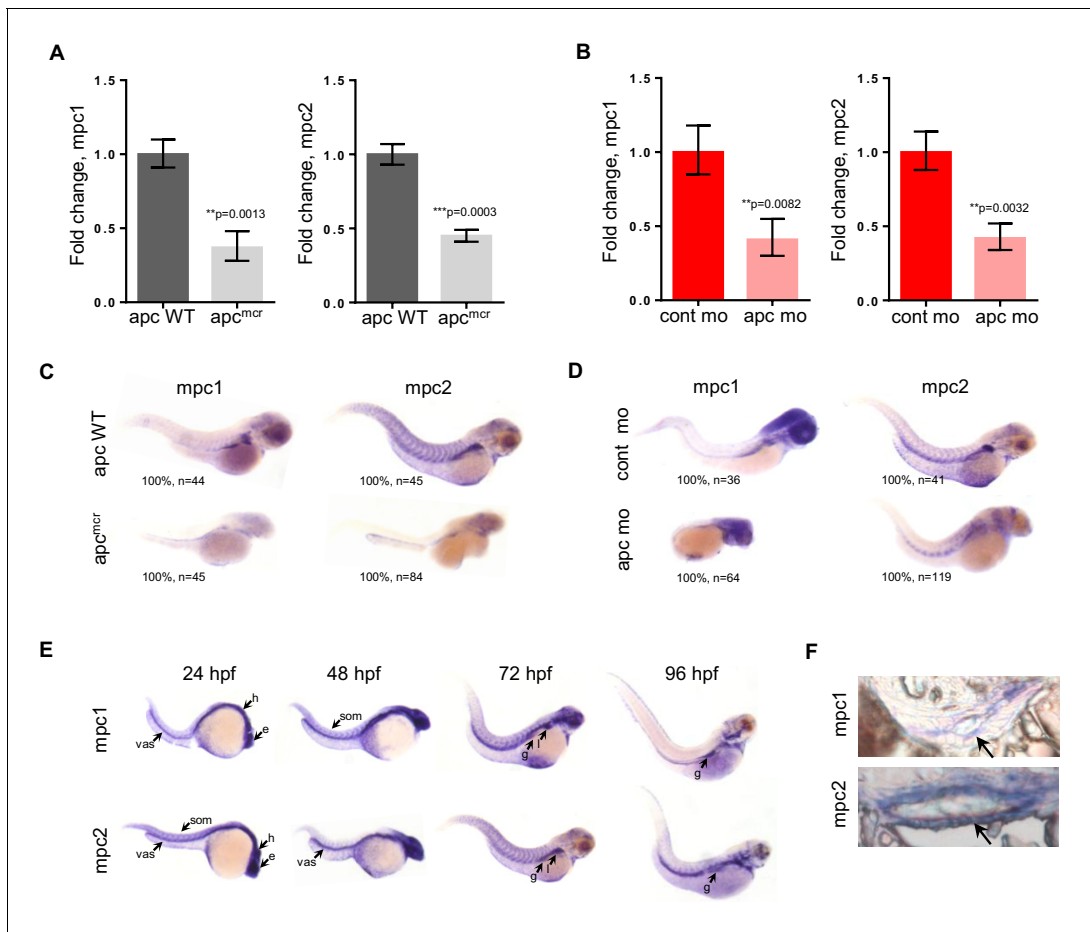

**Figure 1.** *mpc1* and *mpc2* are downregulated in *apc^mcr^* and *apc* morphant embryos. (A,B) Quantitative RT-PCR analysis of *mpc1* and *mpc2* gene expression in *apc^mcr^* (A) and *apc* mo (B) embryos. Values represent mean ± SD. Graph shown above is representative of at least three independent experiments. Statistical significance was analyzed using unpaired t-test. (C,D) Whole mount in situ hybridization for *mpc1* and *mpc2* in 72 hpf *apc^mcr^* (C) and *apc* mo (D) embryos. (E) Whole mount in situ hybridization for *mpc1* and *mpc2* in wild type (WT) embryos. head (h), eyes (e), somite (som), vasculature (vas), gut (g), liver (l). (F) Cross sections from 96 hpf WT embryos probed with either *mpc1* or *mpc2* confirmed gut-specific expression of both genes. See also *Figure 1—figure supplement 1*.

The following source data and figure supplement are available for figure 1:

**Source data 1.** Fold change calculations for *Figure 1A,B*
**Figure supplement 1.** PCR analysis confirming *apc* knockdown.

somites at 24–48 hr post-fertilization (hpf) (*Figure 1E*). At later time points, expression in the pectoral fin buds, liver and gut emerged (*Figure 1E*). Cross sections of 72 hpf WT embryos previously probed with *mpc1* and *mpc2* confirmed gut expression for both genes (black arrows) (*Figure 1F*).

## Knock down of *mpc1* phenocopies *apc* knock down

Previous studies have established phenotypes associated with impaired *apc* function in the developing zebrafish including malformation of the gut, eyes, pancreas and jaw, arrested fin buds and failed heart looping (*Nadauld et al., 2004*; *Hurlstone et al., 2003*; *Nadauld et al., 2006b*). To determine whether loss of *mpc1* would recapitulate morphological defects related to *apc* deficiency, we knocked down the expression of *mpc1* in WT embryos with a splice-blocking morpholino which we confirmed by PCR (*Figure 2—figure supplement 1A*). Microinjection of 0.75 mM *mpc1* morpholino into WT embryos at the one- to two-cell stage resulted in about 87% of injected embryos appearing morphant (n = 228) (*Figure 2—figure supplement 1B*). Consistent with downregulation of *mpc1* in *apc^{mcr}*, *mpc1* morphants (*mpc1* mo) exhibited a range of phenotypes consisting of smaller head and eyes, enlarged hindbrain vesicle (black arrows), pericardial edema, body curvature, and loss of pectoral fins (blue arrows) (*Figure 2A*).

In situ hybridization with *gata6* revealed that the primordial gut formed (93%, n = 55) in *mpc1* mo but developed abnormally, as shown by reduced staining for *fabp2* (100%, n = 36), which marks the differentiated gut (*Figure 2B*). Histological analyses on *mpc1* mo gut confirmed these findings, there were fewer cells comprising the gut tube and they appeared cuboidal and non-polarized (*Figure 2C*). Additionally, intestinal folds were visibly lacking in the *mpc1* mo gut (black arrow, *Figure 2C*). In contrast, cross-section of the gut from control mo showed polarized columnar intestinal cells, with the nuclei lined up clearly against the basal membrane (*Figure 2C*).

Since APC has also been reported to play a crucial role in congenital hypertrophy of retinal pigment epithelium (CHRPE) in humans and normal ocular development in the zebrafish embryo, we examined the eyes of *mpc1* mo and found that *irbp*, a marker for photoreceptor and retinal pigmented epithelial cells, was severely reduced in *mpc1* mo (95%, n = 40) (*Figure 2B*) (*Nadauld et al., 2006b*; *Chapman et al., 1989*). Cross-section of *mpc1* mo eye revealed small lens and disorganized cell layers (*Figure 2C*). The retinal cells appeared to be undifferentiated as supported by the loss of *irbp* expression (red arrow, *Figure 2C*).

Further phenotypic analyses of *mpc1* mo by in situ hybridization using tissue-specific markers exposed diminished maturation for brain (95%, n = 22) and fin buds (100%, n = 28) as indicated by *ascl1a* and *id1* expression, respectively (*Figure 2B*). Also, *mpc1* mo hearts failed to loop as determined by *myl7* staining (100%, n = 39) (*Figure 2B*). As with the gut, terminal differentiation of the pancreas in *mpc1* mo was severely reduced as assessed by *trypsin* expression, a marker for exocrine pancreas (100%, n = 28) (*Figure 2B*). *Insulin*, denoting the endocrine pancreas, remained normal (100%, n = 25) (*Figure 2B*). Cartilage staining with alcian blue confirmed the absence of pectoral fins and revealed improper jaw formation in *mpc1* mo (100%, n = 152) (*Figure 2B*).

We verified that the morphological defects we observed in *mpc1* mo were specifically due to knock down of *mpc1* by co-injecting with 0.5 ng of full length human *MPC1* mRNA and analyzing the embryos by in situ hybridization for *id1* and *otx2*, a marker for both the midbrain and eyes. Overexpression of *MPC1* mRNA alone resulted mostly in normal-appearing embryos, a small percentage exhibited cyclopia, severe body curvature and truncated tail (29%, n = 187) (*Figure 2—figure supplement 1C*). However, in co-injected embryos (*mpc1* mo + *MPC1* RNA), we found that *MPC1* mRNA restored fin development as indicated by *id1* staining (50%, n = 49) (*Figure 2D*, *Figure 2—figure supplement 1D*). We obtained similar results with *otx2*, *MPC1* mRNA was able to rescue normal midbrain and eye development in *mpc1* mo (*Figure 2—figure supplement 1D and E*). The presence of *MPC1* transcript was confirmed by PCR (*Figure 2—figure supplement 1F*).

*mpc1* and *mpc2* form a heterodimer complex that is responsible for transporting pyruvate from the inner mitochondrial space into the inner mitochondrial matrix (*Bricker et al., 2012*; *Herzig et al., 2012*). To examine whether loss of *mpc2* would result in similar phenotypes as *mpc1* mo, we knocked down its expression in WT embryos using antisense morpholino (*Figure 2—figure supplement 2A and B*). We found that *mpc2* mo exhibited similar developmental defects as *mpc1* mo, such as smaller head and eyes, enlarged hindbrain vesicle (black arrows), body curvature, and absence of pectoral fins (blue arrows) (*Figure 2—figure supplement 2B*). In contrast to *mpc1* mo, only a third of *mpc2* mo-injected embryos appeared morphant, the majority of which exhibited a

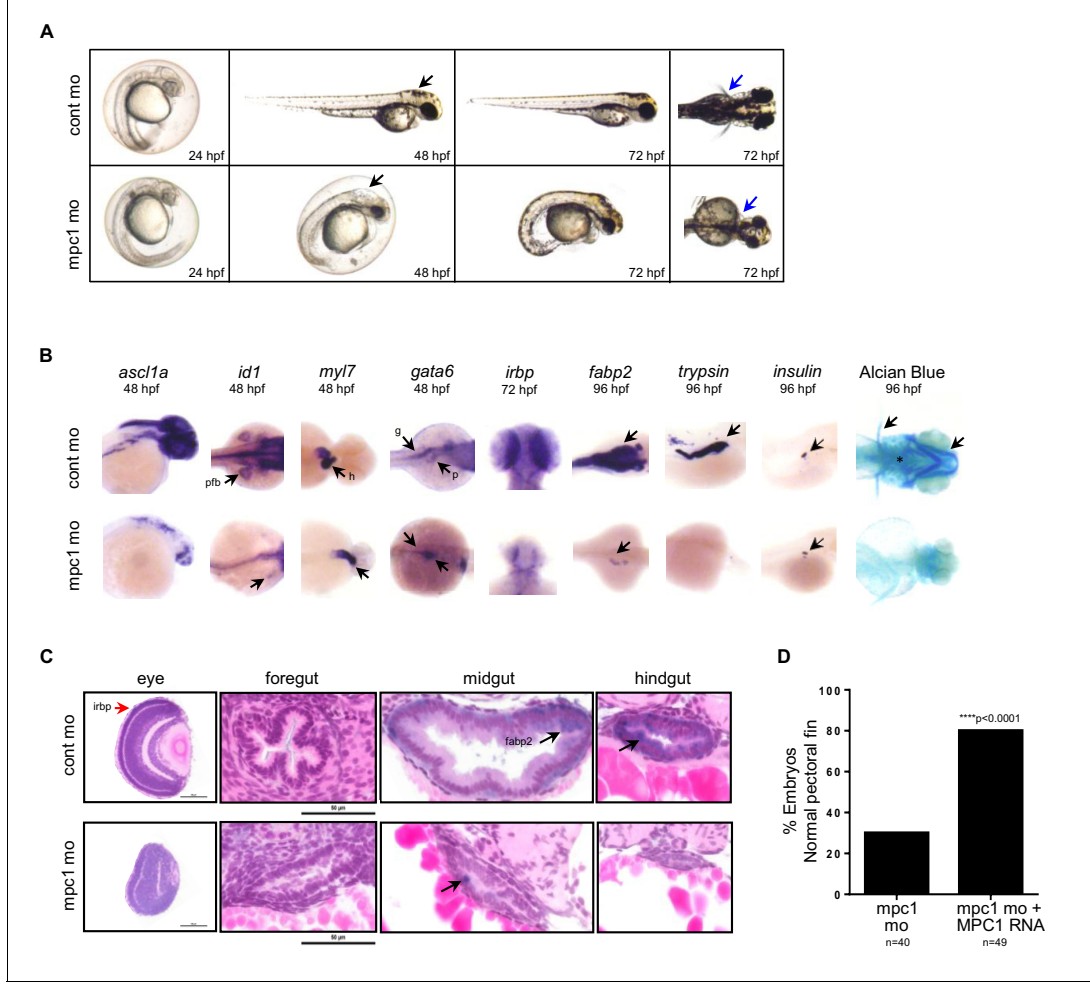

**Figure 2.** Knock down of *mpc1* expression phenocopies loss of *apc*. (**A**) Gross phenotype associated with *mpc1* knock down. (**B**) Whole mount in situ hybridization analysis for organ-specific markers in *mpc1* mo. Alcian blue staining revealed improper cartilage development (*) and confirmed loss of pectoral fins in *mpc1* mo. pectoral fin bud (pfb), heart (h), gut (g), pancreas (p). (**C**) Cross section of the eye and gut from control or *mpc1* mo. Prior to sectioning, embryos were previously stained with eye and gut-specific markers, *irbp* (red arrow) and *fabp2* (black arrow), respectively. (**D**) Co-injection with human *MPC1* RNA led to increased percentage of *mpc1* mo with normal pectoral fins as determined by in situ staining for *idi1*. Statistical significance was analyzed using Fisher's exact test. See also *Figure 2—figure supplements 1, 2*.

The following figure supplements are available for figure 2:

**Figure supplement 1.** *mpc1* morphant phenotype is rescued by human *MPC1* RNA.

**Figure supplement 2.** *mpc2* morphants phenocopy loss of *mpc1*.

mild phenotype (*Figure 2—figure supplement 2C*, *Figure 2—figure supplement 1A*, data not shown). MPC2 expression is inconsistently altered in cancer and variably correlated with survival (*Schell et al., 2014*). We therefore focused further studies on *mpc1*.

### *MPC1* rescues intestinal differentiation in *apc*-deficient embryos

In light of our data relating reduced *mpc1* levels to failed intestinal differentiation, we sought to determine if re-expression of *MPC1* would rescue intestinal defects in *apc*-deficient zebrafish embryos. We injected 0.75 mM *apc* mo with or without 0.1 ng *MPC1* mRNA into WT embryos and evaluated intestinal differentiation by in situ hybridization for *fabp2*. Compared to *apc* mo (6%, n = 265), there was a significant increase in embryos with differentiated gut in the morpholino plus

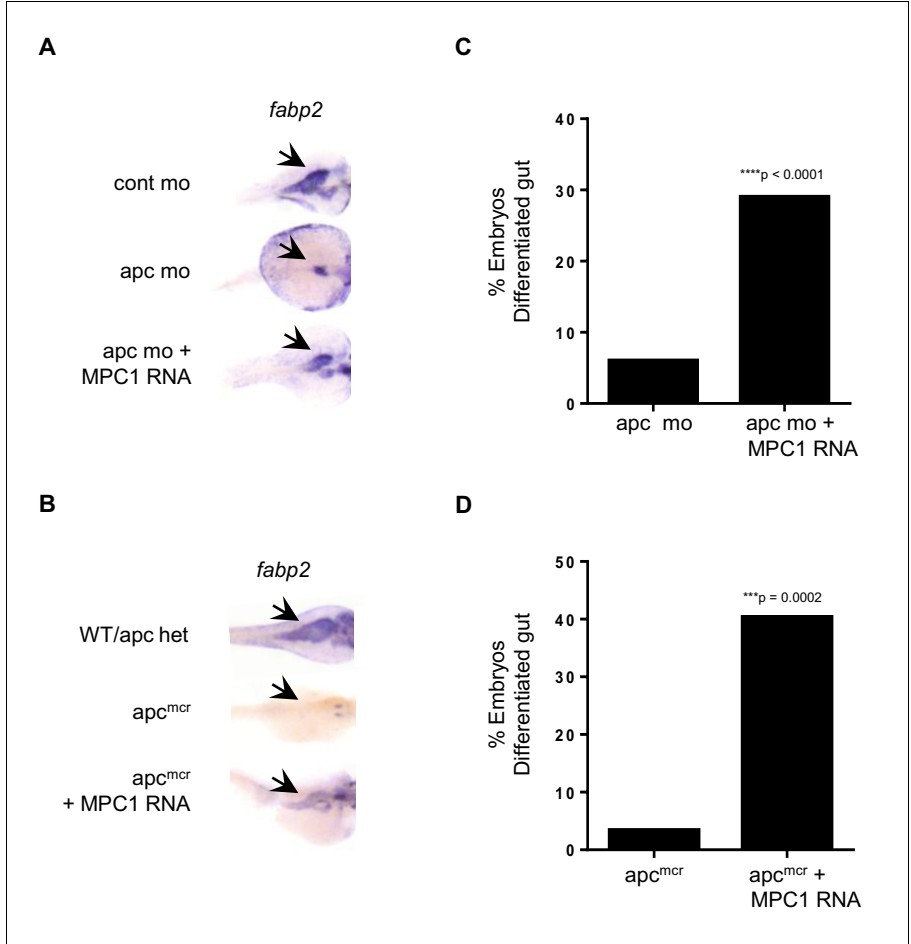

**Figure 3.** *MPC1* rescues gut phenotype of *apc mo* and *apc^{mcr}*. (A,B) In situ hybridization for *fabp2*, in 72 hpf WT embryos injected with cont mo, *apc* mo or both *apc* mo and human *MPC1* RNA (*apc* mo + *MPC1* RNA) (A). In situ hybridization for *fabp2* in 72 hpf *apc* WT, *apc^{mcr}* or *apc^{mcr}* injected with human *MPC1* mRNA (*apc^{mcr}* + *MPC1* RNA) (B). (C,D) Quantification of injected embryos with differentiated gut as determined by *fabp2* staining. Statistical significance was analyzed using Fisher's exact test.

mRNA group (29%, n = 312) (*Figure 3A,C*). Control embryos all displayed normal *fabp2* expression (data not shown).

We confirmed this finding by injecting 0.5 ng *MPC1* mRNA into *apc^{mcr}* and observed similar results, 40% of injected embryos re-expressed *fabp2* (n = 52) (*Figure 3B,D*). However, only 3% of un-injected *apc^{mcr}* showed *fabp2* staining (n = 29), while WT/het siblings were all positive (n = 80, data not shown). *MPC1* was also able to rescue cardiac defects in *apc^{mcr}*, as we saw improved blood circulation in injected mutants as well (data not shown). These results suggest that re-introduction of *mpc1* can drive intestinal differentiation.

## Knock down of *mpc1* or *apc* alters mitochondrial function

Because of the integral role of MPC1 in pyruvate metabolism, we next investigated if *mpc1* mo harbor metabolic defects as a consequence of diminished *mpc1* function. We assessed mitochondrial respiration by measuring oxidative consumption rates (OCR) in 72 hpf embryos and there was a significant reduction in OCR in *mpc1* mo compared to control (*Figure 4A*). We also looked at triglyceride (TG) levels as an indicator of disturbance in normal energy utilization and observed a similar trend (*Figure 4B*). Moreover, there was an extensive dysregulation of pyruvate metabolism upon loss of *mpc1*, as we discovered a profound upregulation of pyruvate metabolic genes in *mpc1* mo, suggesting a compensatory mechanism to account for reduced pyruvate transport across the inner

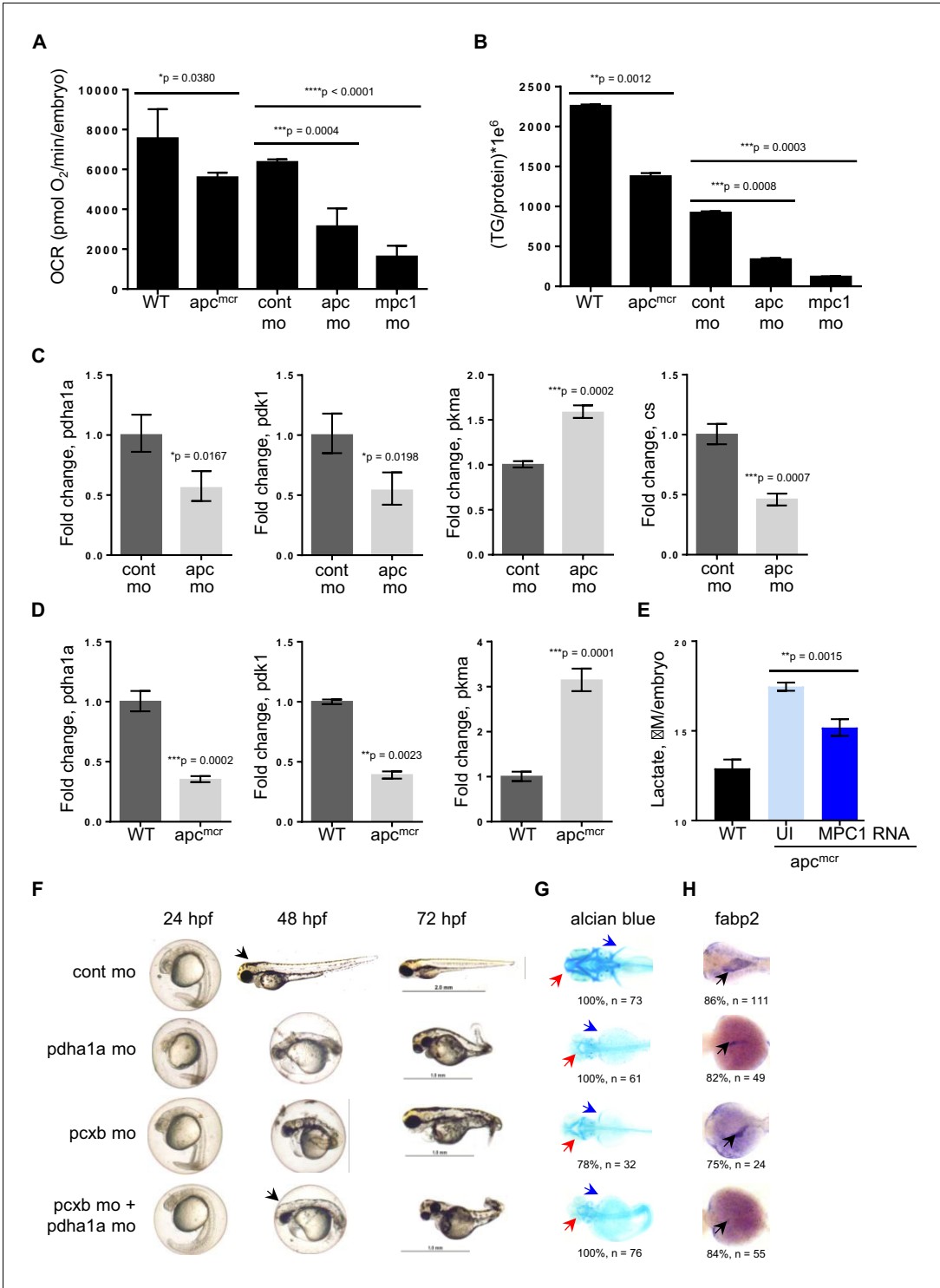

**Figure 4.** Knock down of *mpc1* or *apc* leads to altered mitochondrial respiration and pyruvate metabolism. (**A**) Mitochondrial respiration was evaluated by measuring oxygen consumption rates (OCR) in 72 hpf embryos. (**B**) Triglyceride (TG) levels were determined in lysates prepared from 72 hpf embryos using a colorimetric assay. (**C,D**) Quantitative RT-PCR analysis of enzymes involved in pyruvate metabolism in *apc* mo and (**C**) *apc^mcr* (**D**) embryos. *pyruvate dehydrogenase alpha 1a (pdha1a); pyruvate dehydrogenase kinase, isozyme 1 (pdk1); pyruvate kinase, muscle, a (pkma); citrate synthase (cs).* (**E**) Lactate levels in *apc* wild type (WT), un-injected *apc^mcr* (UI) or *apc^mcr* embryos injected with human *MPC1* mRNA (MPC1 RNA). For figures **A–E**, values represent mean ± SD. Graph shown above is representative of at least three independent experiments. Statistical significance was analyzed

*Figure 4 continued on next page*

*Figure 4 continued*
using unpaired t-test. (F,G,H) Gross phenotype (F), alcian blue staining (G) and in situ hybridization for *fabp2* (H) in *pdha1a*, *pcxb*, and *pcxb* + *pdha1a* mo. *pcxb* (pyruvate carboxylase b). See also ***Figure 4—figure supplements 1***, *2*.
The following source data and figure supplements are available for figure 4:
**Source data 1.** Mean and standard deviation values for ***Figure 4A,B,E***; fold change calculations for ***Figure 4C,D***.
**Figure supplement 1.** Knockdown of *mpc1* leads to dysregulated pyruvate metabolism.
**Figure supplement 1—source data 1.** Fold change calculations for ***Figure 4—figure supplement 1B,C***.
**Figure supplement 2.** PCR analysis confirming knockdown of *pdha1a*, *pcxb*.

mitochondrial membrane (***Figure 4—figure supplement 1A and B***). Knock down of *mpc2* did not affect *mpc1* transcript level (***Figure 4—figure supplement 1C***).

Earlier studies have reported that *MPC1* is downregulated in colon cancer and its expression positively correlates with *APC* (***Schell et al., 2014***). Together with our previous data showing that *mpc1* is downregulated in *apc*-deficient zebrafish and that *mpc1* mo exhibit impaired oxidative respiration, we hypothesized that *apc* regulates *mpc1* and therefore, pyruvate metabolism overall. To test this, we evaluated mitochondrial respiration and TG levels in *apc^mcr* and *apc* mo and compared to appropriate controls, we observed significant defects in mitochondrial function upon loss of *apc* (***Figure 4A,B***). Several enzymes in the pyruvate pathway were also differentially regulated in *apc^mcr* or *apc* mo (***Figure 4C,D***).

To further validate that impaired mitochondrial function in *apc^mcr* is facilitated by reduced *mpc1* expression, we injected *MPC1* mRNA into *apc* mutant embryos and looked at lactate levels as indicator of improved mitochondrial function. Compared to un-injected *apc^mcr*, mutant embryos overexpressing *MPC1* showed a significant reduction in lactate (***Figure 4E***). *apc* WT/het sibs (WT) represent basal lactate levels in normal embryos (***Figure 4E***).

Pyruvate, after passing through the inner mitochondrial membrane, is converted to oxaloacetate and acetyl-CoA by pyruvate carboxylase (PC) and pyruvate dehydrogenase (PDH), respectively. To ascertain if the knock down of enzymes downstream of *mpc1* would result in a phenotype similar to *mpc1* mo, we targeted *pcxb* and *pdha1a*, separately and in tandem, with antisense morpholinos (***Figure 4—figure supplement 2A and B***). Loss of either metabolic gene or both, resulted in morphant embryos that lacked pectoral fins (blue arrows), jaw (red arrows) and appeared identical to *mpc1* mo (***Figures 4F–H***, *2A*). Interestingly, knock down of *pcxb* resulted in only 22% of embryos with phenotype (n = 27) while targeting *pdha1a* gave a higher percentage of morphant embryos (63%, n = 19) (***Figure 4—figure supplement 2C***). A synergistic effect was observed when both enzymes were diminished (83%, n = 18) (***Figure 4—figure supplement 2C***). *fabp2* in situ staining (black arrows) also revealed intestinal developmental defects in the *pdha1a* mo, *pcxb* mo and *pdha1a* + *pcxb* morphant embryos (***Figure 4H***). Knock down of *pcxb* expression, however, not only resulted in low penetrance but mild phenotype as well (***Figure 4F–H***). This could be due to the activation of an alternative pathway where oxaloacetate can be derived from glutamine instead of pyruvate (***DeBerardinis et al., 2007***).

## Decreased RA levels lead to aberrant pyruvate metabolism

To further elucidate how *apc* is controlling *mpc1*, we initially looked at Wnt signaling as one of the major roles of APC is to regulate degradation of β-catenin (***Fearnhead et al., 2001***). Perturbation of the Wnt pathway by treatment of *apc* mo with 10 uM NS-398, a COX-2-specific inhibitor that has been shown to impair β-catenin activity in an *apc*-deficient background, did not affect *mpc1* or *mpc2* levels (***Figure 5A***) (***Eisinger et al., 2007***). We did see a dramatic reduction in expression of known β-catenin target gene, *mmp9*, implying that *apc* regulation of *mpc1* is independent of Wnt (***Figure 5A***).

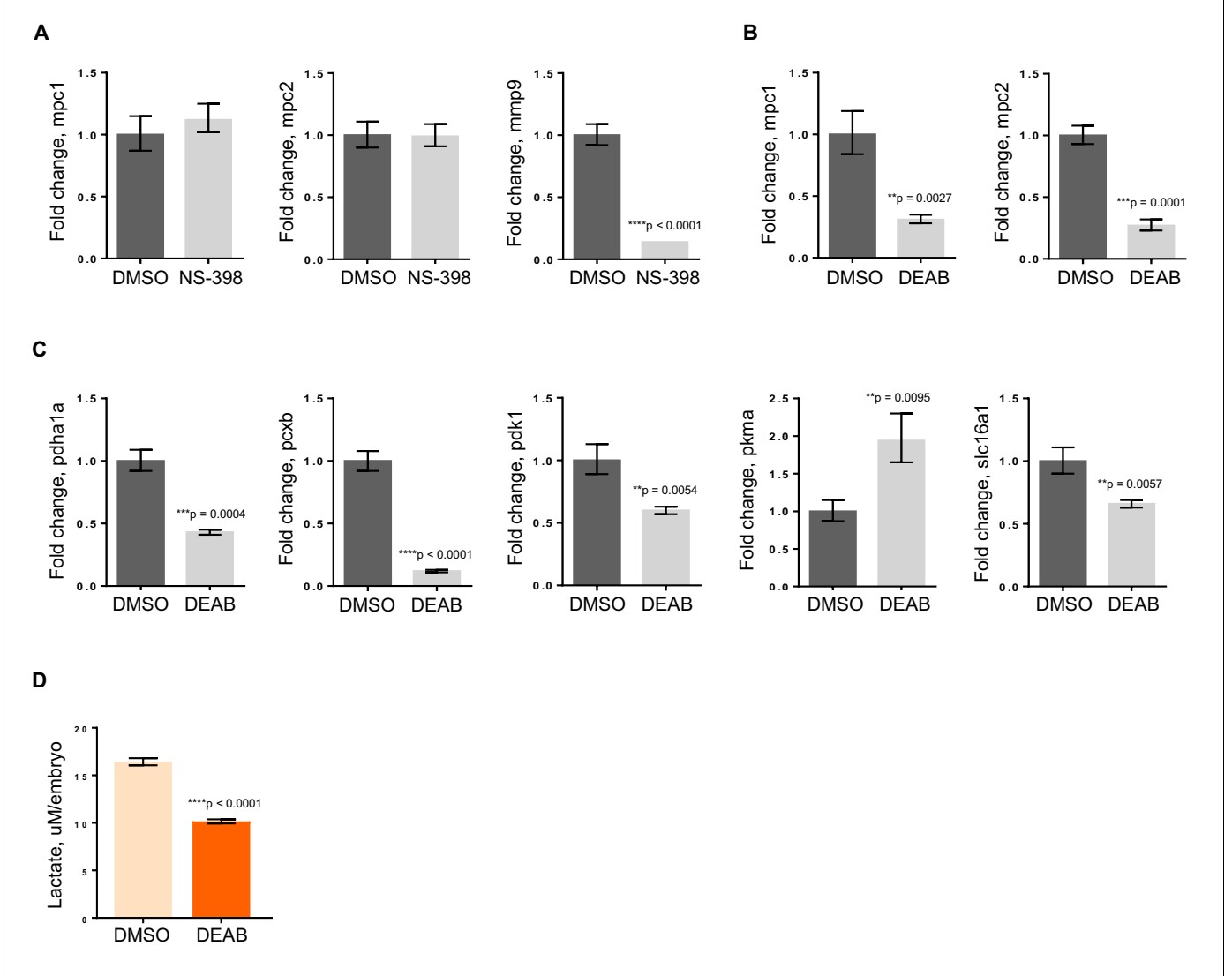

**Figure 5.** RA deficiency results in dysregulated pyruvate metabolism that is independent of Wnt pathway. (**A**) *apc* mo treated either with DMSO control or 10 uM NS-398 were analyzed by qRT-PCR to determine expression levels of *mpc1*, *mpc2* and *mmp9*. (**B,C**) WT embryos treated with either DMSO control or 5 uM DEAB were analyzed by qRT-PCR to determine expression levels of enzymes involved in pyruvate metabolism. (**D**) Lactate levels in 72 hpf WT embryos treated with either DMSO or 5 uM DEAB. For figures **A–D**, values represent mean ± SD. Graph shown above is representative of 3 independent experiments. Statistical significance was analyzed using unpaired t-test. *matrix metallopeptidase 9 (mmp9); solute carrier family 16, member 1 (slc16a1)..*

The following source data is available for figure 5:

**Source data 1.** Fold change calculations for *Figure 5A,B,C*; mean and standard deviation values for *Figure 5D*.

In addition to regulating Wnt signaling, *apc* controls the program of intestinal differentiation through regulation of RA levels, as we have previously shown (*Jette et al., 2004*; *Nadauld et al., 2006a*, *2004*, *2005*; *Rai et al., 2010*; *Shelton et al., 2006*). To interrogate the involvement of RA in pyruvate metabolism downstream of *apc*, we treated WT embryos with DMSO or 5 uM DEAB, a known inhibitor of RALDH which catalyzes the second step in the conversion of Vitamin A to the active metabolite, RA (*Marill et al., 2003*; *Russo et al., 1988*). By qRT-PCR, we found that *mpc1* and *mpc2* transcript levels went down significantly with inhibition of RA (*Figure 5B*). Expanding our analyses to include other components of pyruvate metabolism, there were five other genes that were either up- or downregulated upon DEAB treatment, supporting the notion that the pyruvate

metabolic program is altered at various points when RA levels are perturbed (*Figure 5C*). The DEAB-treated embryos also had low lactate levels, further establishing that RA inhibition results in mitochondrial dysfunction (*Figure 5D*).

## Pyruvate metabolism is dysregulated in colon cancer

To further understand the implication of our findings in zebrafish, we employed publicly available curated databases to investigate mutations and gene expression alterations of pyruvate metabolism genes in human cancers. Using samples deposited at The Cancer Genome Atlas (TCGA), we selected for colon adenocarcinomas with mutations upstream of codon 1600 of APC, a region encompassing the MCR, and resulting in a truncated protein similar to those found in a majority of patients with FAP (*Fearnhead et al., 2001*). Consistent with our previous data, we found that *mpc1* expression is significantly downregulated in colon adenocarcinomas with APC deletions (n = 91) compared to normal colon (n = 19) (*Figure 6A*). We also looked at other genes involved in pyruvate transport and metabolism and interestingly, MPC1, MPC2, PDHA1 and PC showed consistent down-regulation in a specific subset of colon adenocarcinomas known as colon mucinous adenocarcinomas (AC) (n = 22) (*Figure 6—figure supplement 1A*). In addition to altered gene expression, we also found colorectal cancer samples in COSMIC that had mutations both in APC and pyruvate metabolism enzymes. There were five samples with somatic APC deletions that had mutations in multiple pyruvate metabolism enzymes as well, most of which are predicted to be probably (**) or possibly (*) damaging, further supporting a genetic link between APC mutation and dysregulation of pyruvate metabolism (*Figure 6—figure supplement 1B*).

Several studies have shown that individual genes in the pyruvate metabolism pathway are altered in various cancer types (*Gray et al., 2014*; *Schell et al., 2014*). Using Oncomine, we extended these studies by treating the genes involved in pyruvate transport and metabolism as a group (n = 10). We discovered that this pathway is significantly dysregulated in cancer compared to a group of randomly-generated Uniprot genes (n = 55) (*Figure 6B*, *Supplementary file 1*). We then looked at overall survival for patients, with or without mutations and/or gene expression changes in the pyruvate metabolism gene set using TCGA samples in cBioportal. Out of 21 cancer types that we analyzed, only colorectal adenocarcinoma and kidney chromophobe carcinoma showed a significant difference in overall survival between the two groups (*Figure 6C*, *Supplementary file 2*).

## Discussion

Recent identification of *MPC1* and *MPC2*, genes responsible for pyruvate uptake into the mitochondrial matrix, has added a new complexity to targeting pyruvate metabolism in human disorders, including cancer (*Gray et al., 2014*; *Bricker et al., 2012*; *Herzig et al., 2012*). How dysregulation of metabolism relates to the accumulation of genetic hits that cause tumor suppression has largely been unstudied. Here, we demonstrate a direct relationship between loss of a key tumor suppressor gene, *APC*, and dysregulation of *MPC1*. Our finding that *mpc1* expression is downregulated in embryos harboring a genetic mutation (*apc^{mcr}*) or knocked down expression (*apc mo*) of *apc* is reflected in human tumors as well, where we not only discovered a significant downregulation of *MPC1* expression in colon adenocarcinomas with APC deletions, but also samples possessing mutations in both APC and several pyruvate metabolism enzymes (*Figure 1*, *Figure 6*, *Figure 6—figure supplement 1*). Our meta-analyses of human cancer data sets also revealed extensive dysregulation of pyruvate metabolism, at multiple points in the pathway, and this incidence of altered gene expression of pyruvate metabolism genes in cancer is significantly more prevalent compared to a group of randomly selected genes (*Figure 6*, *Figure 6—figure supplement 1*, *Supplementary file 1*). It is remarkable how *MPC1* and *MPC2*, *PDHA1* and *PC*—genes that are involved in the transport and conversion of pyruvate in the inner mitochondrial matrix, respectively—are all significantly downregulated in a subset of colon adenocarcinomas categorized as colon mucinous adenocarcinomas, as loss of these genes essentially shuts down oxidative phosphorylation in the mitochondria (*Figure 6—figure supplement 1*). Interestingly, additional in silico analyses suggest a potential use of pyruvate metabolism genes as prognostic markers for colorectal adenocarcinoma and kidney chromophobe carcinoma, as we found a strong correlation between aberrations in pyruvate metabolism genes with poor overall survival in these cancer types (*Figure 6*).

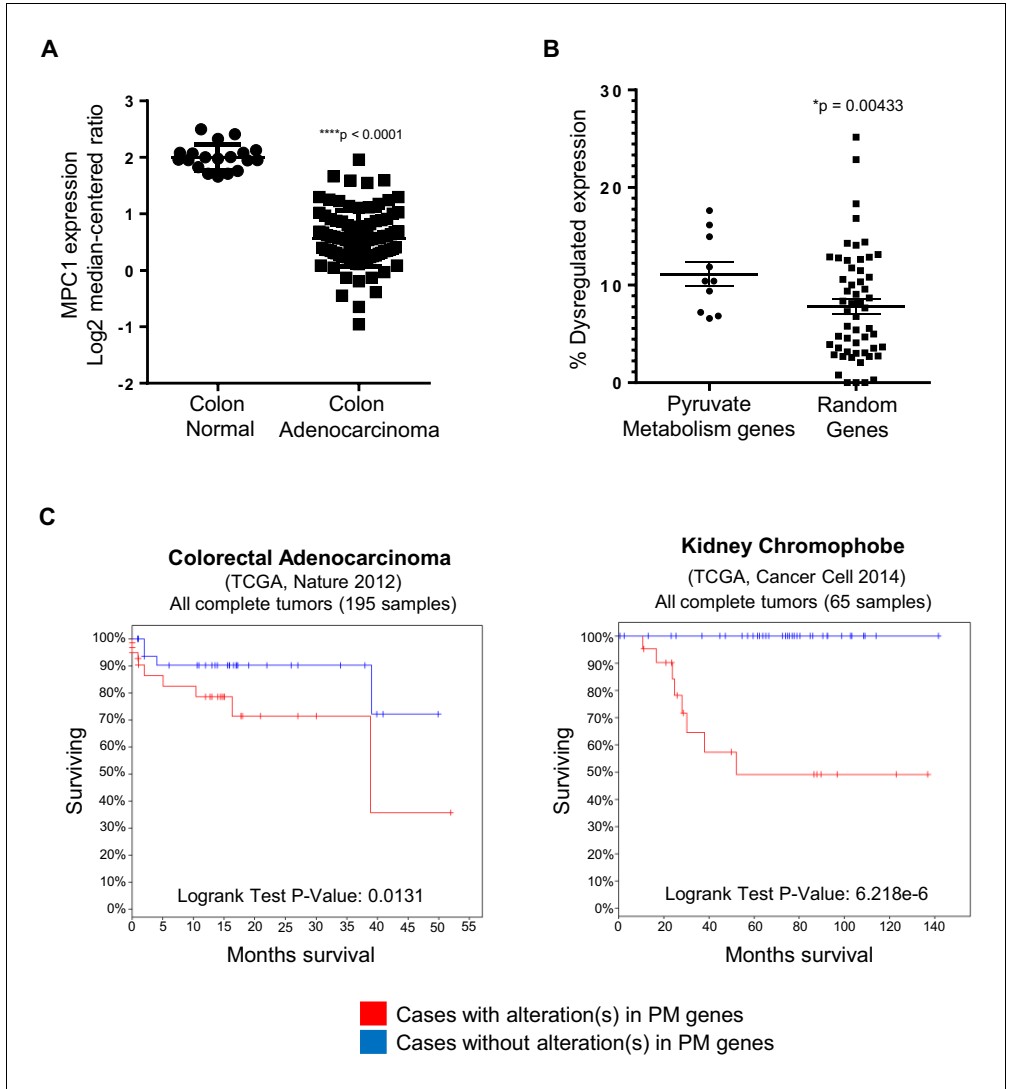

**Figure 6.** In silico analyses of *APC* and pyruvate metabolism gene alterations in cancer. (**A**) *MPC1* expression levels in TCGA normal colon and colon adenocarcinoma samples harboring truncating mutations in APC upstream of codon 1600. Statistical significance was analyzed using Mann Whitney test. (**B**) ONCOMINE database was analyzed for gene expression alterations in pyruvate metabolism genes in all cancer types. A control group composed of Uniprot random genes was used for comparison. Graph shows percentage of up- and downregulated genes with respect to the total unique analyses for each gene tested for all cancer groups. Statistical significance was analyzed using Mann Whitney test. (**C**) cBioportal analysis to estimate Kaplan-Meier overall survival of TCGA colorectal adenocarcinoma and kidney chromophobe patients with or without alterations in pyruvate metabolism (PM) genes. For **B–C**, pyruvate metabolism genes used in meta-analyses: *MPC1*, *MPC2*, *CS*, *PDK1*, *PDHA1*, *PC*, *PKLR*, *LDHA*, *SLC16A1*, *GYS1*. See also *Figure 6—figure supplement 1*, *Supplementary files 1*, *2*.

The following figure supplement is available for figure 6:

**Figure supplement 1.** Pyruvate metabolism genes are mutated in human colon carcinomas.

It is interesting to note that there are differences in the dysregulation of pyruvate metabolism genes in *apc^mcr^/apc mo* and *mpc1 mo* (*Figure 4*, *Figure 4—figure supplement 1*). APC is a multifunctional protein that has critical roles in various cellular processes (*Fodde, 2003*). In addition, regulation of other metabolic genes by APC could occur in parallel with regulation of *mpc1*. *mpc1* knock down alone, therefore, would not alter the expression of these genes in the same way as knock

down of *apc*. Confirming a functional epistatic relationship between *apc* and *mpc1*, knock down of *mpc1* in WT embryos resulted in phenotypes that have been previously reported for impaired *apc* function (*Figure 2*) (*Nadauld et al., 2004*; *Hurlstone et al., 2003*). APC has been shown to positively regulate glycogen synthase kinase-3 (GSK-3) activity, an enzyme that inhibits glycogen synthase (GS) which is involved in converting glucose into glycogen for storage (*Bouskila et al., 2010*; *Valvezan et al., 2012*). Taken together, these findings suggest a major role for APC in controlling cellular bioenergetics and homeostasis, as it can affect glycogen synthesis and oxidative phosphorylation through GSK-3 and MPC1, respectively.

The exact role of metabolism in cancer as a driver versus passenger process has been unclear. Indeed, roles for metabolism in directing cell fate and differentiation decisions are only now being considered (*Schell et al., 2014*; *Bracha et al., 2010*; *Sperber et al., 2015*; *Yanes et al., 2010*; *Zhou et al., 2012*). The rescue of intestinal differentiation defects in embryos with impaired *apc* function by exogenous *MPC1* mRNA establishes a clear role for metabolic programming as a switch that can control cell fate decisions (*Figure 3*). Mechanistic insights into regulatory pathways that control metabolism and how perturbations in cellular bioenergetics effect cell differentiation and proliferation can lead to a better understanding of normal development and tumorigenesis. In this respect, the role of retinoic acid in promoting cell fate and differentiation remains undefined. Our studies suggest that RA may control a program of metabolism that is permissive for intestinal differentiation. The actions of RA are complex, and it is likely that the effects of RA on metabolism are indirect. Consistent with this, treatment of either WT or $apc^{mcr}$ embryos with RA did not result in an immediate induction of *mpc1* (data not shown).

To conclude, we present a novel role for *apc* in controlling a metabolic program driving intestinal differentiation through regulation of *mpc1*. Our data strongly support the notion that metabolic changes are a major part of the decision process in determining cell fate and provide a better understanding of how cancer genetics is linked with biochemical metabolic pathways.

## Materials and methods

### Zebrafish maintenance

Wild-type (WT) *TU* (RRID:ZIRC_ZL57) and $apc^{WT/mcr}$ (RRID:ZFIN_ZDB-ALT-050914-2) Danio *rerio* (zebrafish) were maintained as previously described (*Westerfield, 1995*). Fertilized embryos were collected following natural spawnings in 1 × E3 medium (286 mg/L NaCl, 13 mg/L KCl, 48 mg/L $CaCl_2 \cdot 2H_2O$, 40 mg/L $MgSO_4$, 0.01% methylene blue) and allowed to develop at 28.5°C.

### Morpholino and RNA microinjections

Morpholino oligonucleotides were obtained from Gene Tools LLC (Philomath, OR) and solubilized to 1 mM or 3 mM stock solutions in 1x Danieau buffer. For microinjections, 1 nl of morpholino was injected into WT embryos at the 1- to 2-cell stages (*Draper et al., 2001*). Knock down of gene expression was assessed by PCR. Primers were designed according to guidelines recommended by Gene Tools (www.gene-tools.com) to amplify WT and splice-blocked morphant bands. *z18s* was amplified as a control gene.

For RNA rescue experiments, full length human RNA transcripts were transcribed from linearized DNA using mMESSAGE mMACHINE transcription kit (Ambion - Waltham, MA). For microinjections, 1–2 nl of RNA was injected into embryos at the 1- to 2-cell stages. Overexpression of mRNA transcript was assessed by PCR. Statistical analyses were performed using Fisher's exact test (GraphPad Prism v 6.04, RRID:SCR_002798).

A complete list of morpholinos, PCR primers and working concentrations used are provided in *Supplementary files 3–4*.

### Zebrafish drug treatments

Wild type embryos were given DEAB (VWR International - Radnor, PA) at 5 μM. *apc* morphants were treated with 10 μM NS398 (Cayman Chemical - Ann Arbour, MI). Embryos were harvested at 72 hpf in RNAlater (Ambion) for RNA/cDNA prep.

### In situ hybridization

In situ hybridizations were performed as previously described using digoxigenin-labeled riboprobes for *ascl1a (achaete-scute family bHLH transcription factor 1a)*, *fabp2 (fatty acid binding protein 2, intestinal)*, *gata6 (GATA binding protein 6)*, *id1 (inhibitor of DNA binding 1)*, *insulin*, *irbp (interphotoreceptor retinoid-binding protein)*, *mpc1 (mitochondrial pyruvate carrier 1)*, *mpc2 (mitochondrial pyruvate carrier 2)*, *myl7 (myosin, light chain 7, regulatory)*, *otx2 (orthodenticle homeobox 2)* and *trypsin* (*Thisse and Thisse, 2008*). Embryos were cleared in 2:1 benzyl benzoate/benzyl alcohol solution and documented using an Olympus SZX12/DP71 imaging system (Olympus Corporation - Japan). RNA Reference Sequences deposited in ZFIN (zfin.org, RRID:SCR_002560) were used in designing the riboprobes.

### Quantitative RT-PCR

RNA from zebrafish embryo lysates was isolated using the RNeasy kit (Qiagen - Germany). cDNA was synthesized from 1 µg of total RNA using iScript (Bio-Rad - Hercules, CA). Intron-spanning primers, when possible, were designed using the Universal ProbeLibrary Assay Design Center (Roche Applied Science). A complete list of primer sets is provided in *Supplementary file 5*.

PCR master mix was prepared with the FastStart Essential DNA Probe Master kit and Universal ProbeLibrary probes according to the manufacturer's protocols (Roche Applied Science - Germany). PCR was performed in triplicate using the LightCycler 96 System (Roche Applied Science) with 45 cycles of amplification and annealing temperature of 60°C. Fold change in gene expression was measured by normalizing against 18S rRNA and comparing test group with control.

### Alcian blue assay

Cartilage of 96 hpf embryos was stained with alcian blue as previously described (*Neuhauss et al., 1996*). Briefly, embryos were fixed in 4% sucrose-buffered paraformaldehyde, bleached with 30% hydrogen peroxide for 2 hr and stained with alcian blue overnight. The embryos were then cleared in acidic ethanol for 4 hr, dehydrated stepwise in ethanol and stored either in glycerol or 2:1 benzyl benzoate/benzyl alcohol solution. Stained embryos were examined using an Olympus SZX12/DP71 imaging system (Olympus Corporation).

### Seahorse bioscience XF assay

Metabolic respiration in 72 hpf embryos, expressed as oxygen consumption rate (OCR), was measured using XF24 Extracellular Flux Analyzer (Seahorse Bioscience - North Billerica, MA) as previously described (*Stackley et al., 2011*). As a minor modification, mixing step was omitted during measurement cycle. Statistical analyses were performed using unpaired t-test (GraphPad Prism v 7.02, RRID: SCR_002798).

### Triglyceride (TG) assay

Embryos were harvested at 72 hpf and homogenized in 0.05% PBST +1X protease inhibitor. TG levels were determined using the Infinity Triglycerides Liquid Stable Reagent (Thermo Scientific - Waltham, MA) by measuring absorbance at 540 nm. Total protein concentration was determined using the DC Protein Assay (Bio-Rad) to normalize TG levels. Statistical analyses were performed using unpaired t-test (GraphPad Prism v 7.02, RRID:SCR_002798).

### Lactate assay

Lactate levels were measured in 72 hpf embryos using the EnzyChrom L-Lactate Assay kit (BioAssay Systems - Hayward, CA) as previously described (*Bestman et al., 2015*). Groups of 25–50 embryos were used in the assay. Statistical analyses were performed using unpaired t-test (GraphPad Prism v 7.02, RRID:SCR_002798).

### Histological analyses

Embryos were fixed in 10% neutral buffered formalin, dehydrated in 70% ethanol and embedded in paraffin. Five-micron sections were cut using a Shandon Finesse E Microtome (Thermo Scientific) and stained with hematoxylin and eosin (H and E). Sections were analyzed using a Nikon Eclipse 80i/DS-Fi1 imaging system (Nikon Instruments Inc - Japan).

## Bioinformatic analyses

Publicly available curated databases and analysis software were utilized to examine mutations and gene expression alterations in APC and pyruvate metabolism enzymes (MPC1, MPC2, CS, PDK1, PDHA1, PC, PKLR, LDHA, SLC16A1,GYS1). COSMIC (http://cancer.sanger.ac.uk/cosmic, RRID:SCR_002260) was mined for mutations that are found in human cancers (*Forbes et al., 2015*). Polyphen2 (http://genetics.bwh.harvard.edu/pph2/index.shtml, RRID:SCR_008584) was used to predict functional and structural consequences of amino acid substitutions in proteins mentioned above (*Adzhubei et al., 2010*).

For *MPC1* expression analysis, Oncomine (www.oncomine.org, RRID:SCR_010949) was utilized to determine the colon adenocarcinoma subset (n = 101) in the TCGA (http://cancergenome.nih.gov, RRID:SCR_003193) colorectal carcinoma sample set (n = 237), which was further selected, using COSMIC, for truncating mutations in APC upstream of codon 1600, encompassing the MCR region (n = 91). *MPC1* expression level in these samples was compared with normal colon (n = 19). Colon mucinous adenocarcinomas (n = 22) within the same TCGA colorectal carcinoma sample set were analyzed for gene expression of pyruvate metabolism genes, with normal colon as control. Statistical analysis was performed using Mann Whitney test (GraphPad Prism v 6.04, RRID:SCR_002798).

Oncomine also allowed for gene expression analysis of individual pyruvate metabolism genes (n = 10) and randomly selected Uniprot genes (n = 55) (http://www.uniprot.org/uniprot/?query=reviewed:yes+AND+organism:9606&random=yes, RRID:SCR_002380) in multiple cancer types employing these thresholds: *p value* = 0.001; *fold-change* = 1.5; *gene rank* = top 10%; *data type* = all. A plot was generated to show percentage of datasets meeting set thresholds with respect to total unique analyses for each gene tested in both groups. Statistical analysis was performed using unpaired t-test (GraphPad Prism v 6.04, RRID:SCR_002798).

Overall survival in 21 different TCGA cancer types, segregated by presence of mutations in pyruvate metabolism genes, was analyzed with cBioportal (http://www.cbioportal.org/index.do, RRID:SCR_014555) (*Cerami et al., 2012*). To verify the specificity of our pyruvate metabolism gene set as predictor of overall survival, four groups of ten random genes from Uniprot were utilized as a negative control gene set.

## Statistical analyses

Unpaired t-test was used to compare two unmatched, independent groups. Fisher's exact test was used to determine if outcome is related to a categorical condition by more than chance. Mann-Whitney test was used to compare distribution of two unmatched groups. For fold change data, statistical significance was determined from t-test analyses of relative gene expression. For sample size calculations, the minimum number of samples per group (95% power) was determined by assuming the probability of the defect in the control group is 5% or lower and 80% in the experimental group.

## Acknowledgements

We wish to thank the following: S Lee and HY Lim for assistance with the TG assay, A Vara for technical support, and OMRF Core Labs. This work was supported by NCI/NIH (RO1 CA116468NIH, [DAJ]), Samuel Waxman Cancer Research Foundation, Oklahoma Center for Adult Stem Cell Research (OCASCR) and Oklahoma Medical Research Foundation (OMRF).

## Additional information

### Funding

| Funder | Grant reference number | Author |
|---|---|---|
| National Cancer Institute | RO1 CA116468NIH | David A Jones |
| Samuel Waxman Cancer Research Foundation | | David A Jones |
| Oklahoma Medical Research Foundation | | David A Jones |
| Oklahoma Center for Adult | | David A Jones |

Stem Cell Research

The funders had no role in study design, data collection and interpretation, or the decision to submit the work for publication.

## Author contributions

ITS, Conceptualization, Formal analysis, Investigation, Visualization, Writing—original draft, Project administration, Writing—review and editing; RGCD, Formal analysis, Investigation, Visualization, Writing—original draft, Writing—review and editing; BNM, CS, Investigation, Writing—review and editing; SH, Investigation; KAO, HVR, Resources; AEG, KB, Formal analysis, Investigation, Writing—review and editing; JR, Resources, Writing—review and editing; DAJ, Conceptualization, Supervision, Funding acquisition, Writing—original draft, Writing—review and editing

## Author ORCIDs

Imelda T Sandoval, http://orcid.org/0000-0002-5400-5611
Jared Rutter, http://orcid.org/0000-0002-2710-9765
David A Jones, http://orcid.org/0000-0002-2391-779X

## Ethics

Animal experimentation: This study was performed in strict accordance with the recommendations in the Guide for the Care and Use of Laboratory Animals of the National Institutes of Health. All of the animals were handled according to approved Institutional Animal Care And Use Committee (IACUC) protocol 14-08 of the Oklahoma Medical Research Foundation.

# Additional files

## Supplementary files

• Supplementary file 1. Raw data for oncomine analyses of pyruvate metabolism and random gene dataset expression percent total dysregulation (plotted in *Figure 6B*).

• Supplementary file 2. Overall survival kaplan-meier estimate of pyruvate metabolism geneset alterations using cBioportal (sorted by lowest to highest LogRank Test P-Value).

• Supplementary file 3. List of morpholinos.

• Supplementary file 4. List of PCR primers.

• Supplementary file 5. List of qRT-PCR primers.

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
