## [Decision Letter]

Thank you for submitting your article "A metabolic switch controls intestinal differentiation downstream of Adenomatous Polyposis Coli (APC)" for consideration by *eLife*. Your article has been reviewed by three peer reviewers, and the evaluation has been overseen by Robb Krumlauf as the Senior Editor and Reviewing Editor. The following individual involved in review of your submission has agreed to reveal their identity: Linheng Li (Reviewer #1).

The reviewers have discussed the reviews with one another and the Reviewing Editor has drafted this decision to help you prepare a revised submission.

This is an interesting study, using zebrafish as a model, that describes a link between APC and MPC1 expression in the differentiating gut. The data suggests that APC controls the level of MPC1 through the retinoic acid pathway and shows that knockdown of MPC1 recapitulated phenotypes of impaired APC function including failed intestinal differentiation during development. The study provides insight into the role of APC in regulating pyruvate metabolism in intestinal development. Furthermore, these results may be informative for some of the metabolic effects observed on loss of APC in colon cancer. However, there are a number of concerns that need to be addressed to properly support the conclusions made in the paper and facilitate further consideration for publication in *eLife*.

1) The authors observe that the expression level of several enzymes in the pyruvate pathway are reduced in Apc^mcr^ zebrafish, while the expression levels of the same enzymes increase in MPC1 MO model. This observation raises a question of whether the similar phenotypes of MPC1 KD and APC mutation are the result of effects on the same targets. The authors need to comment on this issue.

2) It is important to show the phenotypes of overexpression MPC1 by injecting mRNA.

3) Did the authors observe any metabolic switch? What's the change of glycolysis and fatty acid metabolism in Apc^mcr^ and MPC1 KD intestines?

4) In Figure 4, the authors show that both APC mutant and MPC1 knockdown zebrafish display reduced OCR. To more rigorously claim that impaired mitochondrial function (OCR) in APC mutants is mediated by reduced MPC1 expression, they could show that OCR is rescued in APC mutants when MPC1 is overexpressed (as they did in Figure 3).

5) If both APC and MPC1 knockdown lead to defects in mitochondrial function, can they reconcile the gene expression differences in TCA cycle enzymes observed with MPC1 knockdown versus APC mutants (Figure 4 vs. Figure 4—figure supplement 1)?

6) An important control is missing to determine whether MPC2 depletion affects MPC1 expression and vice versa.

7) The authors suggest that retinoic acid modulates MPC1 expression yet show little evidence besides gene expression changes to connect retinoic acid to APC and MPC1. This could be very indirect. Are retinoic acid levels decreased in APC mutants? Are metabolic defects (OCR) observed in APC mutants rescued by exogenous retinoic acid? The evidence involving retinoic acid is not very strong.

---

## [Author Response]

[…] 1) The authors observe that the expression level of several enzymes in the pyruvate pathway are reduced in Apc^mcr^ zebrafish, while the expression levels of the same enzymes increase in MPC1 MO model. This observation raises a question of whether the similar phenotypes of MPC1 KD and APC mutation are the result of effects on the same targets. The authors need to comment on this issue.

We agree this needs discussion and have revised the text to point out that APC has multiple functions. In addition, regulation of the various enzymes downstream of APC could occur in parallel with regulation of MPC1. If this were the case, then knockdown of MPC1 would not necessarily result in changes of the other enzymes. The observed difference, as pointed out by the reviewers, suggests this to be the case.

2) It is important to show the phenotypes of overexpression MPC1 by injecting mRNA.

We agree and have provided this as new data in the manuscript (Figure 2—figure supplement 1).

3) Did the authors observe any metabolic switch? What's the change of glycolysis and fatty acid metabolism in Apc^mcr^ and MPC1 KD intestines?

We have observed reduced mitochondrial oxidation and lower triglyceride levels in apc mutants, apc morphants and mpc1 morphants (Figure 4). As an extension, we have also shown that lactate levels are up in apc mutants (Figure 4). We have documented changes in MPC1 levels directly in the intestine by in situ hybridization and observed an intestinal-specific phenotype by knockdown of MPC1 (Figure 1, Figure 2).

4) In Figure 4, the authors show that both APC mutant and MPC1 knockdown zebrafish display reduced OCR. To more rigorously claim that impaired mitochondrial function (OCR) in APC mutants is mediated by reduced MPC1 expression, they could show that OCR is rescued in APC mutants when MPC1 is overexpressed (as they did in Figure 3).

We have performed the requested rescue experiment and have shown that lactate levels are significantly reduced in apc mutants overexpressing MPC1 (Figure 4). We used lactate as a second marker for increased anaerobic glycolysis. This again is consistent with a role for MPC1 downstream of APC.

5) If both APC and MPC1 knockdown lead to defects in mitochondrial function, can they reconcile the gene expression differences in TCA cycle enzymes observed with MPC1 knockdown versus APC mutants (Figure 4 vs. Figure 4—figure supplement 1)?

This is a similar point as number #1 above. Although we think MPC1 and other TCA cycle enzymes are downstream of APC, our data suggest that they are regulated in parallel. MPC1 knockdown alone does not therefore, recapitulate this result.

6) An important control is missing to determine whether MPC2 depletion affects MPC1 expression and vice versa.

We have done this experiment and knockout of MPC2 does not affect levels of MPC1 (Figure 4—figure supplement 1).

7) The authors suggest that retinoic acid modulates MPC1 expression yet show little evidence besides gene expression changes to connect retinoic acid to APC and MPC1. This could be very indirect. Are retinoic acid levels decreased in APC mutants? Are metabolic defects (OCR) observed in APC mutants rescued by exogenous retinoic acid? The evidence involving retinoic acid is not very strong.

A major focus of our lab over the past 12 years is the role of retinoic acid downstream of APC in each of the phenotypes described here (Jette 2004, Nadauld 2004, Nadauld 2005, Nadauld 2006, Rai 2010). Our findings are documented in 12 published manuscripts that illustrated that many of the defects in apc mutant fish, including intestinal differentiation, result from the lack of RA synthesis. This process is controlled mechanistically by CTBP1. We have revised the text to make this point more clear. Although we have firmly established an epistatic role for RA downstream of APC in our previous work, the specific direct functions of RA remain unclear. Our key recent finding suggests that RA effects are indeed indirect by controlling the remodeling of the epigenetic landscape to allow differentiation (Rai 2010). Here we are demonstrating again an epistatic relationship of MPC1 downstream of RA. We have now addressed the direct versus indirect question raised by the reviewers by treating 72 hpf apc mutants with RA. At this time point, we do not observe an immediate induction of MPC1. RA rescue experiments are difficult to perform as RA itself has a profound effect on the development of the embryo overall. We, therefore, rely on RA deficiency experiments to place MPC1 downstream of RA. In this case, blockade of RA synthesis caused phenotypes similar to loss of APC and leads to reduced levels of MPC1. We have revised the manuscript to emphasize the likely indirect regulation of MPC1 by RA.